# Towards Smooth Video Composition

**Qihang Zhang**[1]  **Ceyuan Yang**[2]  **Yujun Shen**[3]  **Yinghao Xu**[1]  **Bolei Zhou**[4]
[1]The Chinese University of Hong Kong,  [2]Shanghai AI Laboratory,  [3]Ant Group,
[4]University of California, Los Angeles

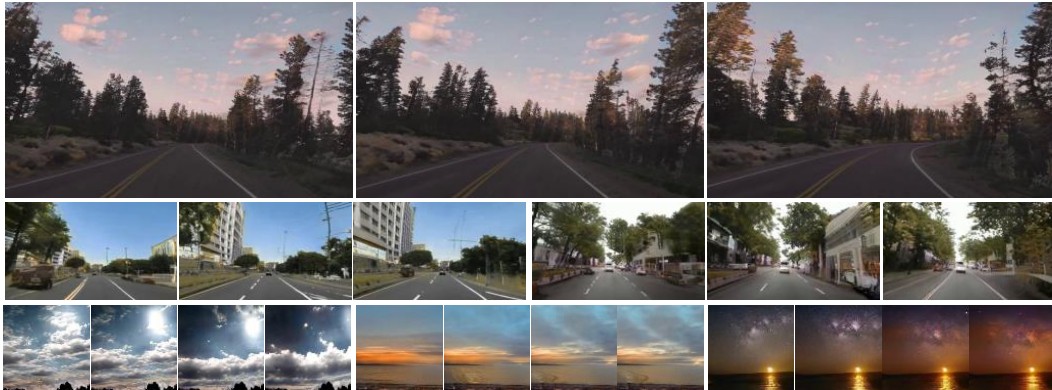

Figure 1: Generated video frames from the proposed model trained on Countryside, YouTube Driving (Zhang et al., 2022), and SkyTimelapse (Xiong et al., 2018) (from top to bottom).

## Abstract

Video generation, with the purpose of producing a sequence of frames, requires synthesizing consistent and persistent dynamic contents over time. This work investigates how to model the temporal relations for composing a video with arbitrary number of frames, from a few to even infinite, using generative adversarial networks (GANs). First, towards composing adjacent frames, we show that the alias-free operation for single image generation, together with adequately pre-learned knowledge, bring a smooth frame transition without harming the per-frame quality. Second, through incorporating a temporal shift module (TSM), which is originally designed for video understanding, into the discriminator, we manage to advance the generator in synthesizing more reasonable dynamics. Third, we develop a novel B-Spline based motion representation to ensure the temporal smoothness, and hence achieve infinite-length video generation, going beyond the frame number used in training. We evaluate our approach on a range of datasets and show substantial improvements over baselines on video generation. Code and models are publicly available at `https://genforce.github.io/StyleSV`.

## 1  Introduction

Synthesizing images using a generative adversarial network (GAN) (Goodfellow et al., 2014; Radford et al., 2016; Karras et al., 2019; 2020b; 2021; 2018) usually requires to compose diverse visual concepts with fine details of a single object and plausible spatial arrangement of different objects. Recent advances in GANs have enabled many appealing applications such as customized editing (Goetschalckx et al., 2019; Shen et al., 2020; Jahanian et al., 2020; Yang et al., 2021) and animation (Qiu et al., 2022; Alaluf et al., 2022). However, employing GANs for video generation remains challenging considering the additional requirement on the temporal dimension.

In fact, a video is not simply a stack of images. Instead, the contents in video frames should have a smooth transition over time, and the video may last arbitrarily long. Thus, compared to image synthesis, the crux of video synthesis lies in modeling the temporal relations across frames. We

argue that the temporal relations fall into three folds regarding the time scale. First, when looking at a transient dynamic, we would focus more on the subtle change between neighbor frames and expect decent local motions, such as facial muscle movement and cloud drifting. Along with the duration getting longer, say a segment, more contents within the frame may vary. Under such a case, learning a consistent global motion is vital. For example, in a video of first-view driving, trees and buildings along the street should move backward together with the car running forward. Finally, for those extremely long videos, the objects inside are not immutable. It therefore requires the motion to be generalizable along the time axis in a continuous and rational sense.

This work targets smooth video composition through modeling multi-scale temporal relations with GANs. First, we confirm that, same as in image synthesis, the texture sticking problem (*i.e.*, some visual concepts are bound to their coordinates) also exists in video generation, interrupting the smooth flow of frame contents. To tackle this obstacle, we borrow the alias-free technique (Karras et al., 2021) from single image generation and preserve the frame quality via appropriate pre-training. Then, to assist the generator in producing reasonable dynamics, we introduce a temporal shift module (TSM) (Lin et al., 2019) into the discriminator as an inductive bias. That way, the discriminator could capture more information from the temporal perspective for real/fake classification, serving as a better guidance to the generator. Furthermore, we observe that the motion representation in previous work (Skorokhodov et al., 2022) suffers from undesired content jittering (see Sec. 2.4 for details) for super-long video generation. We identify the cause of such a phenomenon as the first-order discontinuity when interpolating motion embeddings. Towards this problem, we design a novel B-Spline based motion representation that could soundly and continuously generalize over time. A low-rank strategy is further proposed to alleviate the issue that frame contents may repeat cyclically. We evaluate our approach on various video generation benchmarks, including YouTube driving dataset (Zhang et al., 2022), SkyTimelapse (Xiong et al., 2018), Taichi-HD (Siarohin et al., 2019b) and observe consistent and substantial improvements over existing alternatives. Given its simplicity and efficiency, our approach sets up a simple yet strong baseline for the task of video generation.

## 2 METHOD

We introduce the improvements made on the prior art StyleGAN-V (Skorokhodov et al., 2022) to set a new baseline for video generation. We first introduce the default configuration (**Config-A**) in Sec. 2.1. We then make a comprehensive overhaul on it. Concretely, in Sec. 2.2 we confirm that the alias-free technique (**Config-B**) in single image generation, together with adequately pre-learned knowledge (**Config-C**), could result in a smooth transition of two adjacent frames. Sec. 2.3 shows that when the temporal information is explicitly modeled into the discriminator through the temporal shift module (Lin et al., 2019) (**Config-D**), the generator can produce significantly better dynamic content across frames. Although prior arts could already generate arbitrarily long videos, the cyclic jittering is observed as time goes by. We therefore propose a B-Spline based motion representation (**Config-E**) to ensure the continuity, which together with a low-rank temporal modulation (**Config-F**) could produce much more realistic and natural long videos in Sec. 2.4.

### 2.1 PRELIMINARY

StyleGAN-V (Skorokhodov et al., 2022) introduces continuous motion representations and a holistic discriminator for video generation. Specifically, continuous frames $I_t$ could be obtained by feeding continuous $t$ into a generator $G(\cdot)$:

$$I_t = G(u, v_t), \tag{1}$$

where $u$ and $v_t$ denote the content code and continuous motion representation. Concretely, content code is sampled from a standard gaussian distribution while the motion representation $v_t$ consists of two embeddings: time positional embedding $v_t^{pe}$ and interpolated motion embedding $v_t^{me}$.

To obtain the time positional embedding $v_t^{pe}$, we first randomly sample $N_A$ codes as a set of discrete time anchors $A_i, i \in [0, \cdots, N_A - 1]$ that share equal interval (256 frames in practice). Convolutional operation with 1D kernel is then applied on anchor $A_i$ for temporal modeling, producing the corresponding features $a_i$ with timestamp $t_i$. For an arbitrary continuous $t$, its corresponding interval is first found with the nearest left and right anchor feature $a_l$ and $a_{l+1}$, at time $t_l$ and $t_{l+1}$, so that $t_l \leq t < t_{l+1}, l \in [0, \cdots, N_A - 2]$. For time positional embedding $v_t^{pe}$,

Table 1: **Performance evaluation on various configurations. Config-A** is the baseline model, StyleGAN-V (Skorokhodov et al., 2022). **Config-B** and **Config-C** target fixing the texture sticking issue yet maintain the per-frame quality (Sec. 2.2), which is primarily evaluated by FID. **Config-D** aims to help the generator to produce more reasonable dynamics (Sec. 2.3), which is primarily evaluated by $\text{FVD}_{16}$ and $\text{FVD}_{128}$. **Config-E** and **Config-F** alleviate the discontinuity when interpolating motion embeddings (Sec. 2.4). For all metrics, a lower number is better.

| Configuration | SkyTimelapse | | | YouTube Driving | | | Taichi-HD | | |
| --- | --- | --- | --- | --- | --- | --- | --- | --- | --- |
| | FID | $\text{FVD}_{16}$ | $\text{FVD}_{128}$ | FID | $\text{FVD}_{16}$ | $\text{FVD}_{128}$ | FID | $\text{FVD}_{16}$ | $\text{FVD}_{128}$ |
| **A** StyleGAN-V | **40.8** | 73.9 | 248.3 | 28.3 | 449.8 | 460.6 | 33.8 | 152.0 | 267.3 |
| **B** + Alias free | 54.0 | 118.8 | 221.4 | 56.4 | 729.8 | 886.0 | 31.4 | 171.7 | 522.9 |
| **C** + Image pretrain | 52.2 | 73.5 | 230.3 | **15.6** | 272.8 | 447.5 | **20.8** | 104.3 | 314.2 |
| **D** + TSM | 49.9 | **49.0** | **135.9** | 19.2 | **207.2** | **221.5** | 26.0 | 84.6 | 176.2 |
| **E** + B-Spline | 63.5 | 64.8 | 185.3 | 21.8 | 281.6 | 375.2 | 25.5 | **82.6** | **169.7** |
| **F** + Low-rank | 60.4 | 61.9 | 229.1 | 23.0 | 260.4 | 278.9 | 24.8 | 98.3 | 186.8 |

a wave function is derived with learned frequency $\beta$, phase $\gamma$ and amplitude $\alpha$ from the left anchor feature $a_l$:

$$\alpha = M_\alpha(a_l), \beta = M_\beta(a_l), \gamma = M_\gamma(a_l),$$
$$v_t^{pe} = <\alpha \cdot \sin(\beta \cdot t + \gamma), \alpha \cdot \cos(\beta \cdot t + \gamma)>, \quad (2)$$

where $M_\alpha, M_\beta, M_\gamma$ are learnable MLPs. The interpolated motion embedding $v_t^{me}$ could be obtained through linear interpolation between anchor feature $a_l$ and $a_{l+1}$ based on $t$:

$$v_t^{me} = \frac{(t - t_l)a_l + (t_{l+1} - t)a_{l+1}}{t_{l+1} - t_l}. \quad (3)$$

The final motion representation $v_t$ is the sum of these two terms:

$$v_t = v_t^{pe} + v_t^{me}. \quad (4)$$

Additionally, a sparse sampling strategy is proposed for efficient training. Namely, $N_t$ discrete time steps $t_i$ and real video frames $I_i$ are sampled respectively. By feeding $N_t$ discrete time steps and one content code into generator, multiple synthesis $I_i' = G(u, v_{t_i})$ are obtained. In order to distinguish real video frames from synthesized ones, a discriminator $D(\cdot)$ extracts feature $y_i$ for the individual frame, and then a fusion function $\Theta$ would perform the temporal aggregation over all frames:

$$y = \Theta(y_i) = \Theta(D(I_i)), \ i = 0, 1, \ \cdots, N_t - 1. \quad (5)$$

The final discriminative logit $l$ is computed by a MLP $M$ conditioned on time steps:

$$l = M(y, t_0, \cdots, t_{N_t-1}). \quad (6)$$

In the following context, we would make a comprehensive overhaul on it, leading to a new simple yet strong approach for video generation.

## 2.2 ALIAS-FREE OPERATIONS AND PRE-LEARNED KNOWLEDGE

When composing a few frames, they usually share similar visual concepts and micro motion gradually occurs. Therefore, the smooth transition between frames tends to make the synthesis much more realistic. However, by examining the synthesized frames of StyleGAN-V, we find that texture sticking (Karras et al., 2021) exists. To be specific, we track the pixels at certain coordinates as video moves on and the *brush effect* in Fig. 2a indicates that these pixels actually move little. Thus, the texture sticks to fixed coordinates.

**Alias-free operations (Config-B).** To overcome the texture sticking issue, we follow Style-GAN3 (Karras et al., 2021) to leverage the alias-free technique in the generator, as one could remove the unhealthy dependency on absolute pixel coordinates through decent signal processing. As is shown in Fig. 2b, *brush effect* disappear, *i.e.,* texture sticking is significantly alleviated, with the aid of alias-free technique.

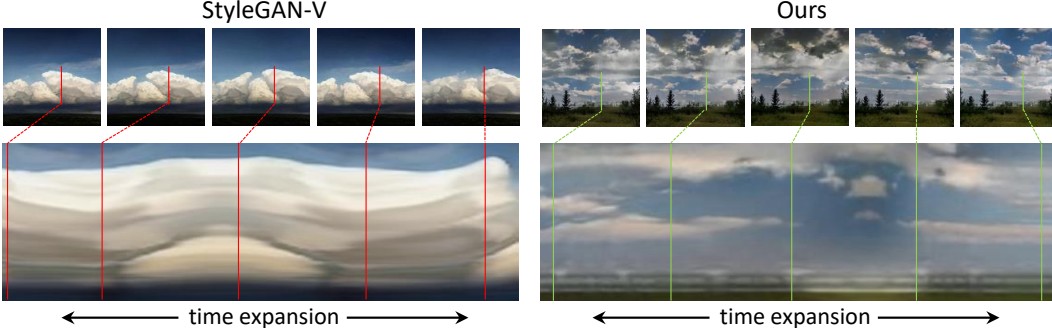

Figure 2: **Visualization of the texture sticking issue.** On the top are a few samples from a continuous sequence of frames, while the bottom visualizes the varying content at a fixed position. We can observe the brush effect from the baseline (Skorokhodov et al., 2022), where the pixel values are strongly bound to the coordinates. Instead, our approach achieves smoother frame transition between neighbor frames.

However, as reported in Tab. 1, using StyleGAN3 achieves lower performance in FID and FVD than before. One possible reason is that the training receipt is directly inherited from the original StyleGAN-V without any modification. Nevertheless, given that it may impair FID and FVD to some extent, we still take StyleGAN3 in for sake of its alias-free benefit.

**Pre-learned knowledge (Config-C).** As simply replacing the generator backbone with StyleGAN3 can deteriorate the synthesis, we propose to make the best of pre-learned knowledge to catch up with the performance. As specified above, the key challenge for video generation is to model temporal relations between frames. However, this target is entangled with the urge to model single image distribution. We confirm that pre-training at image level and then fine-tuning in video level can decouple these two targets well. As discovered in (Karras et al., 2019; Yang et al., 2021), GAN's deep layers mainly control details, for example coloring, detailed refinements, and etc. We hypothesize that these parts' knowledge can be pre-learned and re-used in the video generation learning process.

To be specific, we set $N_t$ as 1 in Eq. (5), and train the whole network from scratch as the image pre-training stage. Then deep layers' weights are loaded in the video fine-tuning stage. With pre-trained deep layers' weights, now the optimization process can focus on early layers to learn to model sequential changes in structures which show natural movements. As shown in Tab. 1, after adopting **Config-C**, the generated quality gets improved compared to **Config-B**. Now, with the alias-free technique, we achieve smooth frame transition without sacrificing much per-frame quality.

### 2.3 EXPLICIT TEMPORAL REASONING IN DISCRIMINATOR

In an adversarial context, it is essential to have a strong discriminator, assuring the sufficient training of the generator. In video generation task, the discriminator thus has to model the temporal relations of multiple frames to distinguish unnatural movements from real ones. However, a simple concatenation operation $\oplus$ is adopted before: $y = \underset{i}{\oplus} y_i$, where $y_i$ denotes a single frame's feature extracted by the discriminator and $y$ denotes the feature after temporal fusion.

**Temporal modeling (Config-D).** We therefore introduce an explicit temporal modeling approach, *i.e.,* temporal shift module (TSM) (Lin et al., 2019) that shows its superiority in standard video understanding area and, most importantly, introduces zero overhead. Concretely, at each convolution layer before temporal fusion, we have features from multiple frames: $y_i^n \in \mathcal{R}^{H \times W \times C}, i = 0, \cdots, N_t - 1$, where $n$ is the layer index, $H$ and $W$ are the feature map resolution and $C$ is the feature channel number. Then, TSM performs channel-wise swapping between adjacent frames:

$$y_i^n(x,y) \leftarrow \oplus_3(y_{i-1}^n(x,y)[: \frac{C}{8}], y_i^n(x,y)[\frac{C}{8} : \frac{7C}{8}], y_{i+1}^n(x,y)[\frac{7C}{8} :]), \quad (7)$$

where $\oplus_3(\cdot)$ is a concatenation operation, and $(x, y)$ are arbitrary coordinates. In this way, one quarter of the channels of a single frame after temporal fusion contain information from

StyleGAN-V

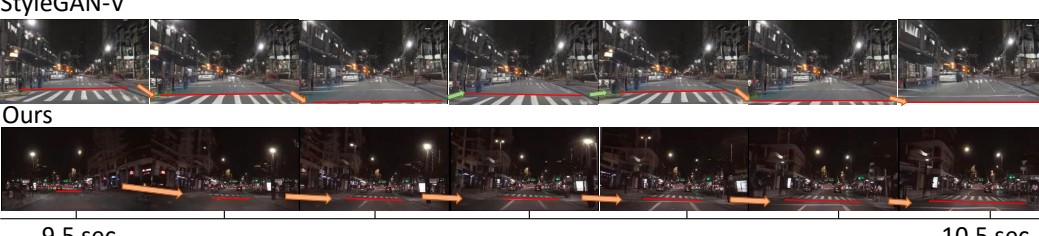

Ours

9.5 sec                                            10.5 sec

Figure 3: **Visualization of the content jittering issue.** As time goes by, we can observe the unstable crossroad in the baseline model (Skorokhodov et al., 2022), caused by discontinuous motion representation. Instead, our approach alleviates such an issue with the proposed B-Spline motion representation, facilitating long video generation.

adjacent frames. Follow-up convolution kernels in deeper layers can perform efficient temporal reasoning based on this mixed representation. In our ablation study, while FID gets worse slightly compared to **Config-C**, adopting explicit temporal modeling (**Config-D**) improves $\text{FVD}_{16}$ and $\text{FVD}_{128}$ significantly, yielding over 100% improvement on YouTube Driving dataset.

## 2.4 TOWARDS UNLIMITED VIDEO GENERATION

Considering that the main metrics *i.e.,* FID, $\text{FVD}_{16}$ and $\text{FVD}_{128}$ used before merely measuring the synthesis of relatively short videos (*e.g.,* 128 frames usually cover around 6 seconds), we further investigate whether current configuration is able to produce arbitrarily long videos. Note that, by saying arbitrarily long, we mention infinite length along time axis, but not infinite content. Ideally, we could easily generate infinite frames by continuously sampling time $t$. The generator is supposed to produce the corresponding synthesis. However, as rollouting the video, a conspicuous jittering phenomenon occurs periodically every 10 seconds. Concretely, as demonstrated in Fig. 3, the crossroad moves forward at the beginning, and then suddenly goes backwards.

**Discontinuity of motion embeddings.** As mentioned in Sec. 2.1, motion embedding $v_t$ contains time positional embedding $v_t^{pe}$ and interpolated motion embedding $v_t^{me}$. In particular, $v_t^{pe}$ could be obtained by a learned wave function on the left nearest anchor feature $a_l$ while the motion embedding $v_t^{me}$ derives from the linear interpolation between the left and right anchor feature $a_l$ and $a_{l+1}$. This interpolation would result in the first-order discontinuity for both learned wave function and linear interpolation when getting through multiple anchors, as shown in Fig. 4a. Moreover, T-SNE (Van der Maaten & Hinton, 2008) is applied to investigate the motion embedding given a long period. Fig. 5a confirms that such discontinuity would cause the drastic change, *i.e.,* jittering content.

**B-Spline based motion representations (Config-E).** With an assumption of the discontinuity of motion embeddings, we design a basic spline (B-Spline) based motion representations that could guarantee first-order numerical smoothness. To be specific, B-Spline of order $n$ is a piece-wise polynomial function of degree $n-1$ in a given variable. It is widely used in computer-aided design and computer graphics for curve fitting, controlling a smooth curve with calculated weights $w_i$ on several control points. In particular, the number of control points is determined by the order of the B-Spline. With a pre-defined knot sequence $\{t_i\}$, a B-Spline with any order can be defined by means of the Cox–de Boor recursion formula (de Boor, 1971). The first-order B-Spline is then defined by:

$$B_{i,1}(t) := \begin{cases} 1 & \text{if } t_i \le t < t_{i+1}, \\ 0 & \text{otherwise.} \end{cases} \tag{8}$$

Here $i$ is the interval index. B-spline of higher orders is further defined recursively:

$$B_{i,k+1}(t) := \omega_{i,k}(t)B_{i,k}(t) + [1 - \omega_{i+1,k}(t)] B_{i+1,k}(t), \tag{9}$$

where $k$ is the order and

$$\omega_{i,k}(t) := \begin{cases} \frac{t-t_i}{t_{i+k}-t_i} & t_{i+k} \ne t_i, \\ 0 & \text{otherwise.} \end{cases} \tag{10}$$

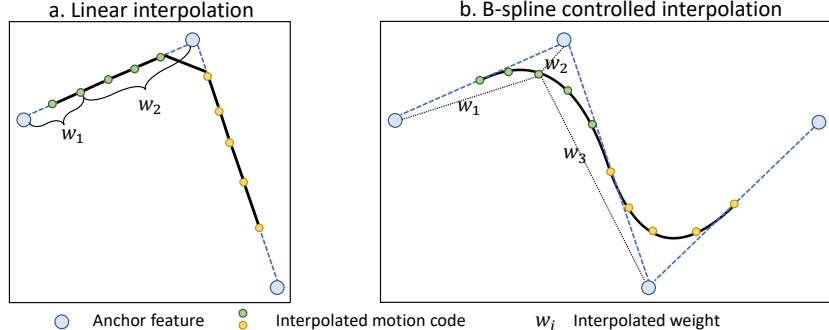

Figure 4: **Comparison between linear interpolation and our B-Spline controlled interpolation.**
B-spline is able to smooth the interpolation between various anchor features, and hence improve the
first-order numerical continuity.

We treat each anchor feature $a_i$ as the control point and use $B_{i,k}(t)$ as its corresponding weight
given time $t$. Hence, B-Spline based anchor feature is defined as:

$$\hat{a}(t) = \sum_i B_{i,k}(t)a_i. \tag{11}$$

We further use $\hat{a}(t)$ instead of discretized $a_i$ to calculate time positional embedding $v_t^{pe}$ and
interpolated motion embedding $v_t^{me}$ as defined in Eq. (2) and Eq. (3). Fig. 4b suggests the continuity
property of B-Spline. Meanwhile, through T-SNE again, the B-Spline based motion representations
become much smoother in Fig. 5b. As shown in Fig. 3, the crossroad gradually approaches the
observer without any sudden jittering.

**Low-rank temporal modulation (Config-F).** After adopting B-Spline based motion embedding,
the jittering phenomenon gets erased. However, we find that similar content periodically appears
as time goes by (see Fig. A6a). This phenomenon implies that motion embedding represents visual
concepts in an ill manner. We thus hypothesize that the new motion embedding might be endowed
with a stronger capacity to represent various visual concepts. Inspired by recent works (Bau et al.,
2020; Wang et al., 2022a), we suppress the representation capacity through the low-rank strategy.

Original StyleGAN-based architecture incorporates styles into convolutional kernels via modulation
trick (Karras et al., 2020a). When generating videos, a similar technique is also applied with small
modifications. Content embedding $u$ together with the motion embedding $v_t$ would be fed into an
affine layer $M$, generating the style code to modulate the original weight $W$:

$$W' = W \cdot M(u \oplus v_t), \tag{12}$$

where $\oplus$ stands for the concatenation operation. That is, motion and content embedding could
equally contribute to the final style to change the visual concepts of frames. In order to suppress the
representational capacity of motion embeddings, we first separate the motion and content codes and
then replace the kernel regarding the motion code with a low-rank one:

$$W' = W_{co} \cdot M_{co}(u) + W_{mo} \cdot M_{mo}(v_t), \text{ where } W_{mo} = UV, \tag{13}$$

where $U, V$ is tall matrix that guarantees $W_{mo}$ is a low-rank matrix. With such a low-rank strategy,
the capacity of motion embedding is suppressed. The repeating issue is alleviated in Fig. A6b.

In conclusion, to generate a short video (less than 10 seconds), **Config-D** is the best choice. In
contrast, one can use **Config-F** for long video generation.

## 3 EXPERIMENTS

We evaluate our method on multiple datasets and compare it against prior arts. Additionally, some
good properties are also given to support the effectiveness of our approach.

**Evaluation Metrics.** Akin to previous literature, Frechet Inception Distance (FID) (Heusel et al.,
2017) and Frechet Video Distance (FVD) (Unterthiner et al., 2018) serve as the quantitative metrics

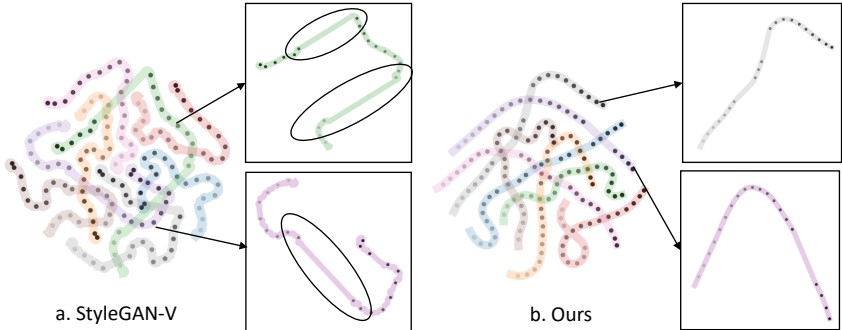

Figure 5: **Visualization of motion embeddings** using T-SNE (Van der Maaten & Hinton, 2008). Each dot refers to a motion code $v_t$, which represents a single frame's motion. A set of progressively darkening dots construct one trajectory that represents a synthesized frame sequence. We can tell that there usually exist undesired twists in the motion trajectory of StyleGAN-V (Skorokhodov et al., 2022), indicating the discontinuity of motion embedding. On the contrary, our approach helps alleviate this issue, resulting in smoother transition throughout the entire video.

Table 2: **Quantitative comparison** between our approach and existing video generation methods on SkyTimelapse (Xiong et al., 2018). For all metrics, a smaller number is better.

| Method | FID | $\text{FVD}_{16}$ | $\text{FVD}_{128}$ |
|---|---|---|---|
| MoCoGAN (Tulyakov et al., 2018) | - | 206.6 | 575.9 |
| + StyleGAN2 backbone | - | 85.9 | 272.8 |
| MoCoGAN-HD (Tian et al., 2021) | - | 164.1 | 878.1 |
| VideoGPT (Yan et al., 2021) | - | 222.7 | - |
| DIGAN (Yu et al., 2022) | - | 83.1 | 196.7 |
| LongVideoGAN (Brooks et al., 2022) | - | 116.5 | 152.7 |
| TATS-base (Ge et al., 2022) | - | 132.6 | - |
| StyleGAN-V (Skorokhodov et al., 2022) | **40.8** | 73.9 | 248.3 |
| Ours | 49.9 | **49.0** | **135.9** |

to evaluate the synthesis for image and video respectively. In terms of FVD, two temporal spans (covering consecutive 16 and 128 frames) are chosen to measure the synthesized videos.

**Benchmarks.** In Sec. 2, we have already conducted comprehensive ablation studies on SkyTimelapse (Xiong et al., 2018), Taichi-HD (Siarohin et al., 2019b), and YouTube Driving dataset (Zhang et al., 2022). To be specific, SkyTimelapse contains over two thousand videos which in average last twenty seconds. Various types of scenes (daytime, nightfall, and aurora) are included. Similar to StyleGAN-V (Skorokhodov et al., 2022), we resize the videos of SkyTimelapse to $256 \times 256$ resolution. Taichi-HD contains over three thousand videos recording a person performing Taichi. We also resize the videos to $256 \times 256$ resolution. YouTube Driving dataset (Zhang et al., 2022) consists of 134 first-view driving videos with a total length of over 120 hours, covering up to 68 cities, showing various conditions, from different weather, different regions, to diverse scene types. The driving scenes have tight geometrical constraints, with most of the objects (vehicles, buildings) following the rigid body constraint. We resize the videos to $180 \times 320$ resolution. A part of videos with countryside views in $320 \times 640$ resolution is also chosen to benchmark high resolution video generation. For training, we resize them to $256 \times 256$ and $512 \times 512$ resolution respectively. Additionally, we compare our approach against previous methods including MoCoGAN (Tulyakov et al., 2018) and its StyleGAN2 based variant, MoCoGAN-HD (Tian et al., 2021), VideoGPT (Yan et al., 2021), DIGAN (Yu et al., 2022), and StyleGAN-V (Skorokhodov et al., 2022).

**Training.** We follow the training receipt of StyleGAN-V and train models on a server with 8 A100 GPUs. In terms of various methods and datasets, we grid search the $R_1$ regularization weight, whose details are available in Appendix. In particular, performances of MoCoGAN and its StyleGAN2 based variant, MoCoGAN-HD, VideoGPT, and DIGAN are directly borrowed from StyleGAN-V.

Table 3: **Quantitative comparison** between our approach and existing video generation methods on Taichi-HD (Siarohin et al., 2019b). For all metrics, a smaller number is better.

| Method | FID | $FVD_{16}$ | $FVD_{128}$ |
|---|---|---|---|
| MoCoGAN-HD (Tian et al., 2021) | - | 144.7 | - |
| DIGAN (Yu et al., 2022) | - | 128.1 | - |
| TATS-base (Ge et al., 2022) | - | 94.6 | - |
| StyleGAN-V (Skorokhodov et al., 2022) | 33.8 | 152.0 | 267.3 |
| Ours | **26.0** | **84.6** | **176.2** |

Table 4: **Quantitative comparison** between our approach and StyleGAN-V (Skorokhodov et al., 2022) on YouTube Driving dataset (Zhang et al., 2022). For all metrics, a smaller number is better.

| Resolution | Method | FID | $FVD_{16}$ | $FVD_{128}$ |
|---|---|---|---|---|
| 128 | StyleGAN-V | 28.3 | 449.8 | 460.6 |
| | Ours | **19.2** | **207.2** | **221.5** |
| 512 | StyleGAN-V | 14.6 | 262.4 | 285.4 |
| | Ours | **14.5** | **116.0** | **139.2** |

## 3.1 MAIN RESULTS

As shown in Tab. 2, our method significantly outperforms existing baselines on SkyTimelapse in terms of $FVD_{16}$ and $FVD_{128}$. On Taichi-HD, our method achieves consistent improvement in terms of FID, $FVD_{16}$, and $FVD_{128}$ as reported in Tab. 3. We also compare our method with StyleGAN-V in challenging YouTube Driving dataset. at both $256 \times 256$ and $512 \times 512$ resolution. For $256 \times 256$ resolution in Tab. 4, our method achieves better performance in both FID, $FVD_{16}$, and $FVD_{128}$. For $512 \times 512$ resolution, StyleGAN-V has a comparable FID score but much worse FVD score. For both two resolutions on YouTube driving, we yield a strong performance of less than 50% FVD score than the prior art. As shown in Fig. 1, our method achieves an appealing visual effect, with smooth frame transition.

## 3.2 PROPERTIES

**Good translation equivariance.** We adopt alias-free technique from image generation for a smooth frame transition. This design thus allows for geometric manipulation of the video with strong translation equivariance using a human-specified translation matrix. We showcase several examples in Fig. 6 by feeding different translation matrix to the video generator. It is clear that videos can be successfully translated, and translation equivariance is well maintained over time. This equivariance property improves human controllability over the video generation process.

**No jittering phenomenon.** As mentioned in Sec. 2.4, discontinuity exists in the motion representation in the prior method, around which jittering phenomenon appears in generated videos. It is caused by discretized anchor features with fixed interval lengths (256 frames in practice). Our method develops a novel B-Spline based motion representation to ensure the temporal smoothness over the anchor

Table 5: FVD evaluated on the frames around anchors.

| | $FVD_{16}$ | $FVD_{16}$-anchor |
|---|---|---|
| **Config-D** | 207.2 | 269.1 (+61.9) |
| **Config-F** | 260.4 | 250.2 (-10.2) |

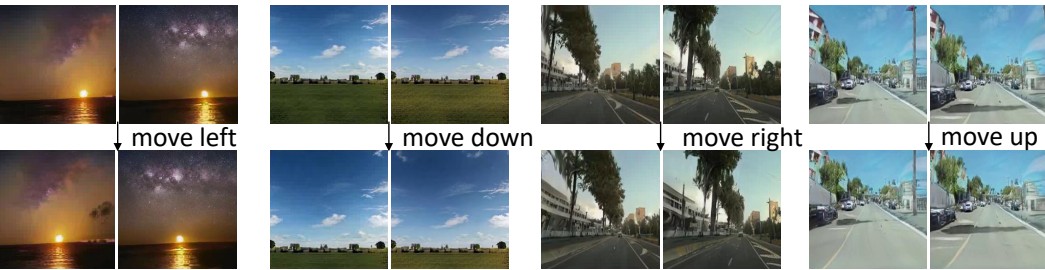

Figure 6: **Application of video 2D translation**, where we are able to control the shift of the frame content, such as moving the setting sun left and moving the grassland down.

features. To quantify the smoothness of generated videos, we sample 16 frames across the anchor (eight frames before and behind the anchor) and calculate the $FID_{16}$ metric (which we name "$FID_{16}$-anchor") in YouTube Driving dataset. As reported in Tab. 5, **Config-D** suffers from severe performance drop, while **Config-F** with B-Spline based motion representation maintains the performance.

## 4 RELATED WORK

**Synthesizing videos from latent space.** Most of the video generation frameworks are built upon GANs (Goodfellow et al., 2014; Brock et al., 2019; Karras et al., 2019; 2020b; 2021; 2018; Radford et al., 2016), owing to their flexibility as well as strong performance in image generation. Prior arts (Tian et al., 2021; Saito et al., 2017; Tulyakov et al., 2018; Fox et al., 2021; Skorokhodov et al., 2022; Yu et al., 2022; Vondrick et al., 2016; Saito et al., 2020; Clark et al., 2019; Wang et al., 2022b) adopt image generators to synthesize frames from a time-coherent sequence of latent codes generated by recurrent networks. To explicitly disentangle the motion from the content, MoCoGAN (Tulyakov et al., 2018) and TGAN (Saito et al., 2017) employ a motion code and a content code as input noise for the generator, serving as a common strategy for the later works. StyleGAN-V (Skorokhodov et al., 2022), which we use as a strong baseline, and DIGAN (Yu et al., 2022) both use implicit neural-based representations for continuous video synthesis. Brooks et al. (2022) leverages a multi-resolution training strategy to prioritize the time axis and ensure long-term consistency. Another line of works (Yan et al., 2021; Kalchbrenner et al., 2017; Weissenborn et al., 2019; Rakhimov et al., 2020; Walker et al., 2021) use autoregressive models to achieve video generation. Kalchbrenner et al. (2017) use a PixelCNN (Van den Oord et al., 2016) decoder to synthesize the next frame pixel by pixel in an autoregressive manner. VideoGPT (Yan et al., 2021) built upon VQ-VAE adopts an autoregressive model to decompose videos into a sequence of tokens. TATS-base (Ge et al., 2022) similarly leverages 3D-VQGAN to decompose videos into tokens and further uses transformers to model the relationship between frame tokens. Additionally, some works using diffusion-based models also present promising results on video generation. Video diffusion (Ho et al., 2022) models entire videos using a 3D video architecture while Yang et al. (2022) uses an image diffusion model to synthesize frames with a recurrent network.

**Synthesizing videos with guided conditions.** A close line of research is video prediction, aiming to generate full videos from the observation of previous frames (Babaeizadeh et al., 2017; Kumar et al., 2019; Lee et al., 2018; Luc et al., 2020; Nash et al., 2022; Finn et al., 2016) or a given conditions (Kim et al., 2020; 2021; Ha & Schmidhuber, 2018). These works typically employ the reconstruction losses to make the future frames predictable for the model. Video translation (Pan et al., 2019; Yang et al., 2018; Wang et al., 2018; Chan et al., 2019; Siarohin et al., 2019a; Ren et al., 2020; Siarohin et al., 2019b) is also a paradigm for video synthesis, which translates the given segmentation masks and keypoints into videos. Some (Li et al., 2022; Liu et al., 2021; Ren & Wang, 2022) focus on synthesizing videos of indoor and outdoor scenes for given camera trajectories, which need to model the 3D scene explicitly. Besides, Ho et al. (2022); Hong et al. (2022); Wu et al. (2021a;b) explore text-to-video generation with diffusion-based models and transformers.

## 5 CONCLUSION

In this work, we set up a simple yet effective framework for video generation task. By introducing the alias-free operations together with pre-learned knowledge, per-frame quality is improved. Explicit temporal modeling in discriminator enhances the dynamic synthesis. In addition, a new B-Spline based motion representation with low-rank temporal modulation is proposed to improve infinite frame generation. Experimental results show substantial improvements on different datasets.

**Discussion.** Our approach has several limitations. Our current framework uses two latent codes to represent content and motion, but they often entangle with each other. Moreover, the frame quality is still far from satisfactory. The object shape is not consistent across consecutive frames, producing apparent artifacts. This could be further improved by introducing structural priors. Last, although we could generate infinite videos by gradually changing content code, direct control of novel content over time is lacking. Explicit planning in the latent space, as used by Ge et al. (2022), could improve the quality of generated long videos.

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

# APPENDIX

This appendix is organized as follows. appendix A and appendix B present the implementation and dataset details respectively, followed by appendix D showing more visual results.

## A  IMPLEMENTATION DETAILS.

Our method is developed based on the official implementation of StyleGAN-V (Skorokhodov et al., 2022). We adopt hyper-parameters, the optimizer, the loss function, and the training script to ensure a fair comparison. Notably, the total number of seen images for sufficient data training is 25 million regardless of datasets. We evaluate once after training on every 2.5 million images, and report the result with highest $FVD_{16}$ score.

Due to statistics variance of different datasets, we search training regularization term, *i.e.* $R_1$ value, for each method and dataset. Empirically, we find that a smaller $R_1$ value (*e.g.,* 0.25) works well for pretraining stage (**Config-C**). While a larger $R_1$ value (*e.g.,* 4) better suits to video generation learning.

## B  DATASET DETAILS.

In this section, We introduce the datasets we use, briefly analyze their characteristics and challenges brought for video generation, and show sampled real video frames.

**SkyTimelapse.** SkyTimelapse (Xiong et al., 2018) is a time-lapse dataset collected from the Internet showing dynamic sky scenes, such as the cloudy sky with moving clouds, and the starry sky with moving stars. It contains various conditions, for instance, different weather conditions (daytime, nightfall, dawn, starry night and aurora), different kinds of foreground object (cloudy sky, starry sky, and sunny sky), and different motion patterns of the sky. It also contains some other objects like trees, mountains, and buildings which further improve the visual diversity. However, since clouds are fluids, there are very few constraints on movement. Minor disturbances and deformations can make a generated video look realistic.

The whole dataset contains over two thousand videos which in average last twenty seconds. We resize the videos to $256 \times 256$ following prior works. We sample several video clips in Fig. A1.

**YouTube Driving.** YouTube Driving Dataset (Zhang et al., 2022) is crawled from YouTube which contains a massive amount of real-world driving frames with various conditions, from different weather, different regions, to diverse scene types. To be specific, 134 videos with a total length of over 120 hours are collected, covering up to 68 cities. Fig. A2 shows sampled frames. The videos are resized to $256 \times 256$ resolution for training.

The driving scenes have tighter geometrical constraints, with most of the objects (vehicles, buildings) following the rigid body constraint. The 3D structure of the whole scene should be maintained without any deformation. Besides, different types of objects show different motion patterns. Roads and buildings recede at a reasonable speed opposite to that of ego car, while other cars show lane changes, stops, accelerations.

**Countryside.** We select a part of videos with countryside scenes from YouTube Driving dataset and resize them to $512 \times 512$ resolution for training. We use this dataset to benchmark the performance of generating high-resolution videos. Real samples are presented in Fig. A3.

**Taichi-HD.** Taichi-HD (Siarohin et al., 2019b) is a collection of YouTube videos recording a single person performing Tai-chi. It contains 3049 training videos in total, with various foreground objects (people with different styles of clothing) and diverse background. It requires realistic human motion generation, which contains non-rigid motion modeling, which poses a significant challenge for video generation.

Table A1: **User study result.** We report the percentage of favorite users for the quality of a set of single images, a set of short videos, and a set of long videos under models from three different settings.

| | YouTube Driving | | | Taichi-HD | | |
|---|---|---|---|---|---|---|
| | image(%) | short video(%) | long video(%) | image(%) | short video(%) | long video(%) |
| Config-A | 4 | 6 | 8 | 0 | 2 | 4 |
| Config-D | **62** | **60** | 12 | **52** | **66** | 36 |
| Config-F | 34 | 34 | **80** | 48 | 32 | **60** |

## C  USER STUDY

To further prove our method's efficacy, we conduct user study over baseline (**Config-A**), **Config-D**, and **Config-F** on YouTube Driving dataset and Taichi-HD dataset. We evaluate from three perspectives:

**Single Image Quality.** We provide user with a collection of images randomly sampled from generated videos.

**Short Video Quality.** We provide user with a collection of generated videos with consecutive 128 frames (up to about 5 seconds).

**Long Video Quality.** We provide user with a collection of generated videos with consecutive 500 frames (up to 20 seconds).

As shown in Tab. A1, in terms of single image quality, users prefer **Config-D** over other settings in both YouTube Driving and Taichi-HD dataset. The same trend occurs in short video quality assessment. With pre-learned knowledge and a stronger discriminator with temporal shift module (TSM), **Config-D** reaps a significantly higher number of votes. However, for long video generation, **Config-F** becomes the best choice for both YouTube Driving and Taichi-HD dataset. Unnatural jittering and discontinuous motion can drastically affect the users' video viewing experience. With B-Spline based motion embedding and low rank time modulation, **Config-F** is undoubtedly the best option for long video generation.

## D  MORE VISUAL RESULTS.

**Texture Sticking phenomenon.** Texture sticking exists in videos generated by prior methods. Our method leverage the alias-free technique in the generator to overcome this issue. Fig. A4 shows more comparison between our method and StyleGAN-V (Skorokhodov et al., 2022).

**Jittering phenomenon.** Motion embedding designed in previous method is not first-order continuous, leading to jittering phenomenon when generating long videos. We develop B-Spline based motion embedding with smooth property which alleviates the problem. We show examples comparing video sequences generated by our method with StyleGAN-V (Skorokhodov et al., 2022) in Fig. A5. Our videos are smooth, without abnormal movements. In videos generated by StyleGAN-V (Skorokhodov et al., 2022), street lights and zebra crossings may suddenly change their direction of movement.

**Repeating contents.** With B-Spline based motion embedding, **Config-E** can compose smooth long videos. However, similar contents appear periodically as shown in Fig. A6a. B-Spline based motion embedding is endowed with so strong capacity that it represents content and motion concepts simultaneously rather than motion concepts only. We suppress the representation capacity through the low-rank strategy. As shown in Fig. A6b, **Config-F** with low rank time modulation can generate progressively forward contents without the repeating phenomenon.

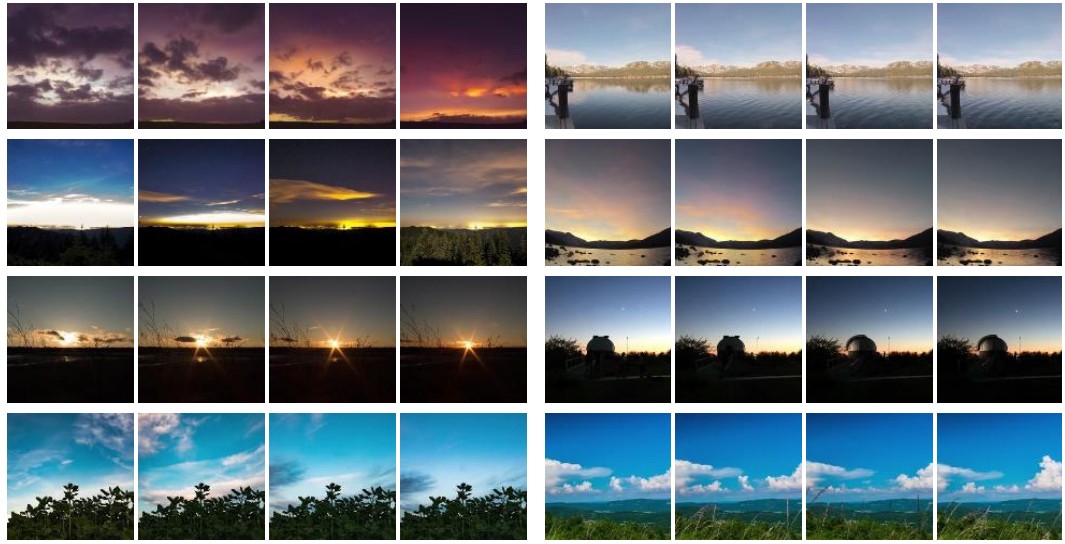

Figure A1: Sampled video frames from SkyTimelapse (Xiong et al., 2018).

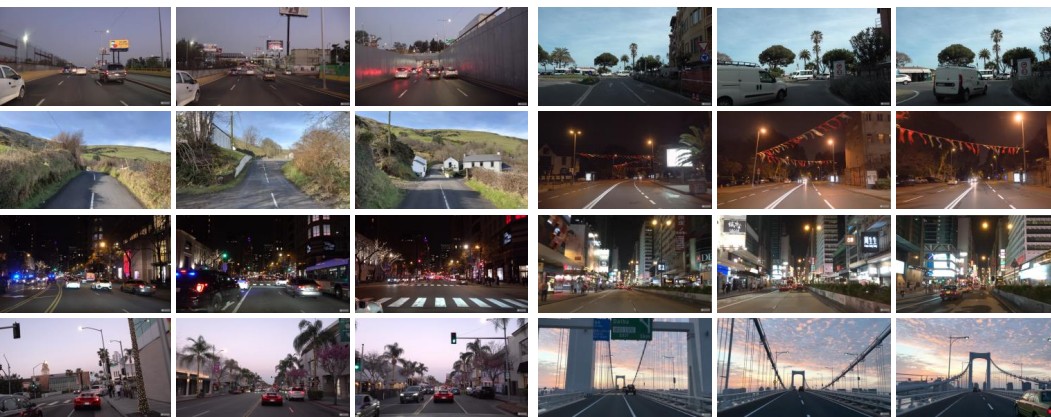

Figure A2: Sampled video frames from self-collected YouTube Driving dataset.

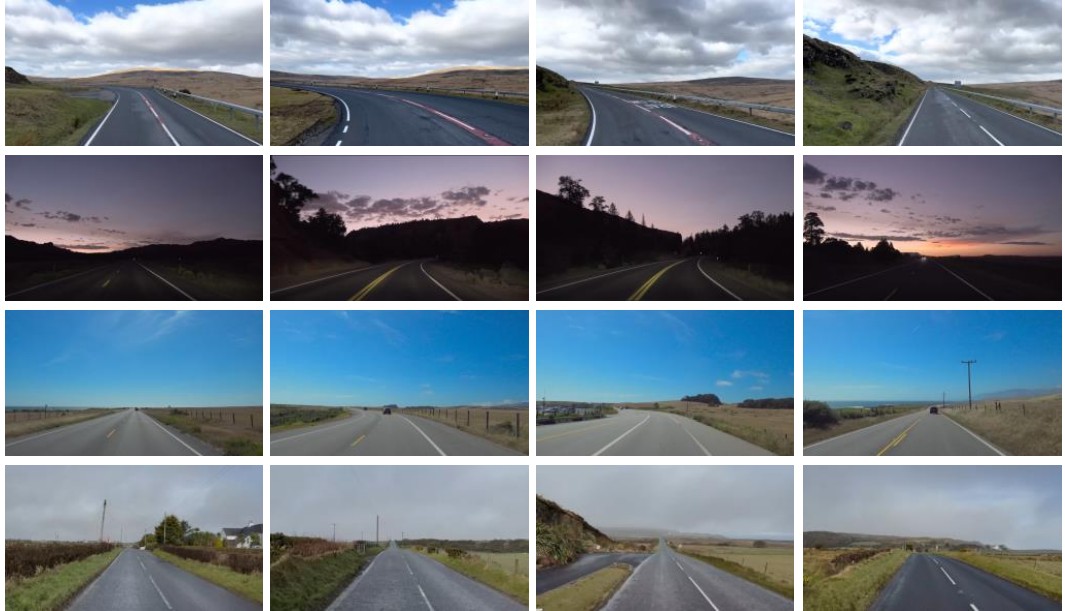

Figure A3: Sampled video frames from self-collected Countryside dataset.

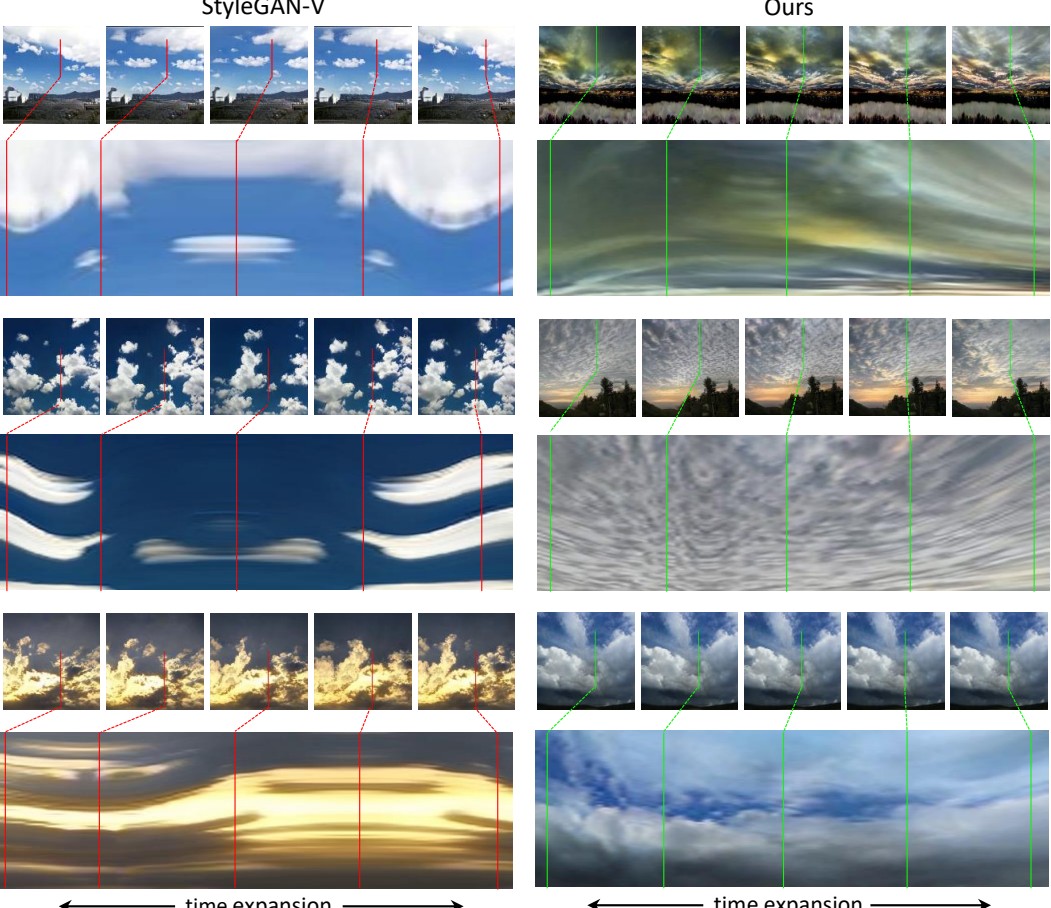

Figure A4: **More examples on texture sticking phenomenon.** We concatenate slices in each frame of the generated video (marked by red and green). We observe *brush effect* when texture sticks at specific location. Instead, our approach achieves more smooth frame transition between neighbor frames.

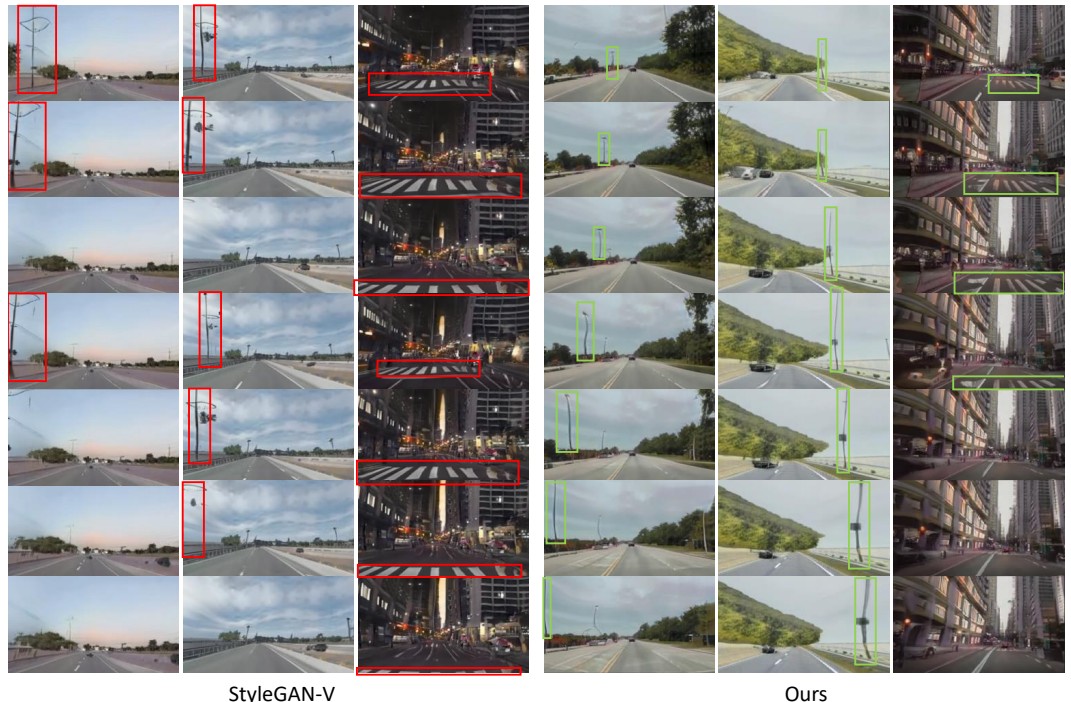

StyleGAN-V                                                                    Ours

Figure A5: **More examples on jittering phenomenon.** Left three columns denote video frames produced by StyleGAN-V (Skorokhodov et al., 2022). Obviously, as highlighted in red bounding boxes, the street lights and crossroad move unsteadily (*e.g.,* gradually fading out first and suddenly fading in). On the contrary, our approach enables much more reasonable and continuous dynamics of certain objects (see green bounding boxes).

a. Config-E                                          b. Config-F

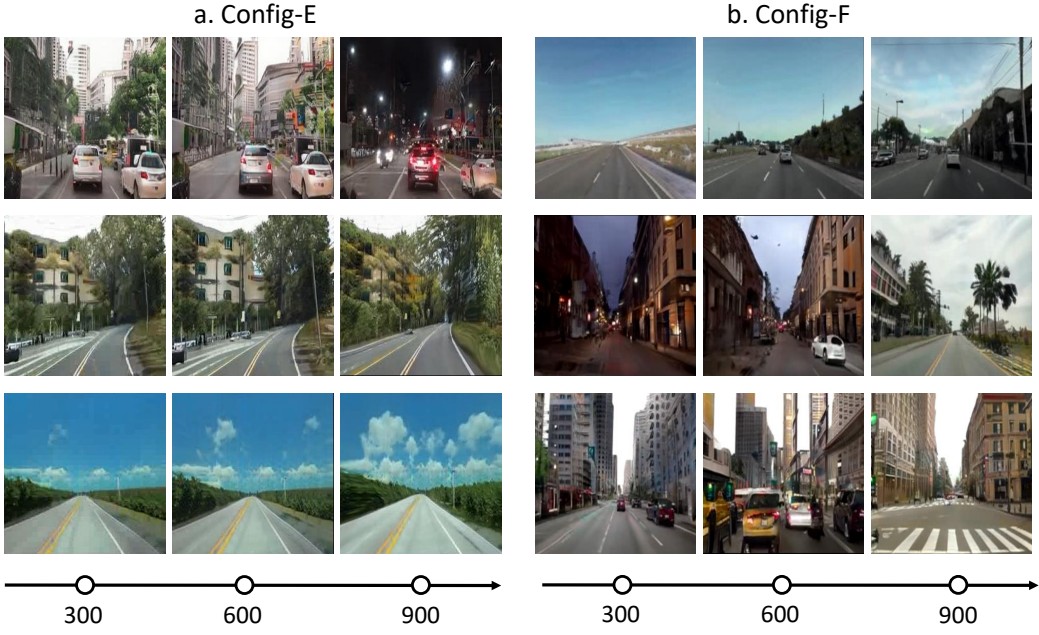

Figure A6: **Examples on repeating contents.** Left three columns denote video frames produced by **Config-E** at the 300th, 600th, and 900th frame respectively. Similar contents (such as foreground objects, and scene layout) periodically appear. In contrast, after adopting low rank time modulation (**Config-F**), contents change gradually without repeating.

