# OpenReview forum: "Towards Smooth Video Composition"
_ICLR.cc/2023/Conference — ICLR 2023 poster_

### Official Review · Reviewer_U7vD · 2022-10-16

**Confidence:** 4
**Correctness:** 3
**Technical Novelty And Significance:** 3
**Empirical Novelty And Significance:** 3
**Recommendation:** 6

**Clarity, Quality, Novelty And Reproducibility:**

The novelty and clarity are fine, evaluation is ok. I guess I'm a bit underwhelmed by the result quality in the longer videos, though.

**Strength And Weaknesses:**

Strengths:
- Clear contributions
- New ideas are explained well

Weaknesses:
- Prior art description (Sec 2.1) is confusing. What data is used to compute the anchor frame embeddings? I suspect the current description cannot be understood by anyone who doesn't already know how it works.
- The result quality is still pretty low -- don't get me wrong, it's better than StyleGAN-V, but we definitely still see jittering in the longer videos and content warps around like crazy. These needs to be admitted and discussed in the paper.
- "Infinite" video is a dubious concept. If the authors want to talk about it, then it needs to be defined. What constitutes an infinite video and what does not? Surely any interesting "infinite" video will need to introduce infinite amount of novel content over time? Otherwise it's hardly infinite -- although just ping ponging a short clip may technically be infinite, but that's hardly a productive definition.



**Summary Of The Paper:**

This paper deals with generative modelling of videos, and specifically proposes a variety of improvements over StyleGAN-V. The improvements are intellectually appealing, and quite convincingly demonstrated. While Arxiv is starting to be full of video papers, many of which exceed the quality of this paper, they may not strictly speaking constitute prior art as even the NeurIPS decisions are not officially out yet. As such, I'm ok with the scientific novelty in this one.

**Summary Of The Review:**

This paper describes a clear step over StyleGAN-V, and as such I'm in favour of acceptance. Details are listed above. Obviously we need better quality metrics for video. To the best of my knowledge Sec. 2.2--2.4 are novel when compared to officially published work, but I may not know everything.

---

> ### Author Response · Authors · 2022-11-15
> **Response to Reviewer U7vD**
>
> **Q1: Prior art description is confusing.**
>
> A1:  The anchor frame embeddings are not pre-computed, but instead randomly sampled and further processed by Conv1D operation. We *have clarified this in Sec. 2.1*.
>
> **Q2: The result quality is still pretty low. These need to be admitted and discussed in the paper.**
>
> A2: Thanks. We *have added a detailed discussion about the limitations and future work in Sec. 5*. You are correct that, even outperforming existing alternatives by a clear margin, the video quality in this work is still far from satisfactory. Some potential solutions are introducing structural priors, applying explicit motion supervisions, improving per-frame generation with local discrimination, *etc.*
>
> **Q3: “Infinite” videos should be defined.**
>
> A3: By saying “infinite”, we would like to refer to a video sequence with a sufficiently large number of frames. In other words, we focus more on reasonably extending the generated video along the time axis (such that it may even go beyond the video length in the training set) instead of generating infinite visual concepts, which is limited by the dataset. We *have added the definition in Sec. 2.4* to make this clear.

---

### Official Review · Reviewer_yrRK · 2022-10-23

**Confidence:** 4
**Correctness:** 3
**Technical Novelty And Significance:** 2
**Empirical Novelty And Significance:** 3
**Recommendation:** 8

**Clarity, Quality, Novelty And Reproducibility:**

[Clarity]

This paper is generally well-written and easy to follow. I have some questions regarding the method details:

1. When pretraining the model on images, what is input as the motion embedding?
2. When loading the pretrained parameters to the model, are the shallow layers initialized and trained from scratch?
3. When generating infinite length of video, what happens if $t$ is sampled to be larger than $N_A-1$?
4. Which layers are TSM added? The notations are a bit confusing in section 2.3 since I previously thought that $y_i$ are the output of the discriminator convolution layers and input to the MLP layers. But it seems that from the top of page 5, more convolution layers are applied to $y_i$?

[Quality]

As stated in the weaknesses, I have several concerns about the experiments done in the paper.

[Novelty]

The proposed methods are incremental but relatively novel in their current forms compared with existing methods.

[Reproducibility]

The code is not provided as part of the supplementary material. I have raised a few questions regarding the details of the method in the clarity section, which could make the method less reproducible.


**Strength And Weaknesses:**

[Strengths]

1. The paper is very well-organized. The author gave a distinct description of the baseline StyleGAN-V model and clear discussions about the problems that existed in this model and how they propose to fix them.

2. The motivations for adopting the temporal shift module (TSM) in the discriminator and pretraining the model on images first are reasonable, adapting techniques proposed in StyleGAN3 to video generation is also useful for the community. The experiments showing improvements brought by these modifications are relatively sound.

[Weaknesses]

There are several flaws in the experiments of this paper:

1. Only two landscape video datasets are used to evaluate the method. It is often the case that people are also interested in generating videos with human motions, which could imply different difficulties from landscapes. There are common video generation benchmark datasets for that purpose such as UCF-101, Tahich-HD, or FaceForensics where previous methods are evaluated on. However, none of these datasets is used in the paper. In addition, since the I3D model used to compute FVD scores is trained on the Kinetics dataset, I personally think that this metric will be more meaningful on the human motion datasets.

2. Even though only two datasets are considered, the trends of the results on them are not well aligned. For instance, pretraining model on images helps on the driving dataset but not the sky dataset, while using TSM only improves the sky dataset. Is this because of the large variance? Probably more datasets could also help demystify these observations.

3. For the main results, the authors adopt only those baselines used in the StyleGAN-V paper. However, since some of the results are focused on long video synthesis, the recently published methods on this problem should be considered [1, 2].

4. The most novel part of the method contains configurations E and F. The main goal of these two fixes is to resolve a known issue of StyleGAN-V which generates cyclic videos when extending to a longer duration. However, these fixes don't seem to improve the model in terms of quantitative evaluation. In addition, only a limited number of qualitative results (in the format of images) are given to demonstrate the improvements. Therefore, the claimed improvements are not convincing to me. Probably some metrics to detect the recurrence or human evaluation could better show the advantage of the proposed method.

In addition, the contribution of the proposed YouTube Driving dataset is not significate to me. There are multiple existing video datasets for driving or car dash cameras, e.g. BDD-100K [3] or Cityscapes[4]. There are also video generation works that are trained on these datasets. The paper lacks a comparison of the collected driving dataset with these existing datasets.

[1] Long video generation with time-agnostic vqgan and time-sensitive transformer. ECCV 2022.

[2] Generating long videos of dynamic scenes. NeurIPS 2022.

[3] Bdd100k: A diverse driving dataset for heterogeneous multitask learning. CVPR 2020.

[4] The cityscapes dataset for semantic urban scene understanding. CVPR 2016.

**Summary Of The Paper:**

This paper improves the previous video generation model StyleGAN-V with several existing techniques and model-specific fixes. The proposed method shows better FVDs on the Sky Time-lapse dataset and a self-collected Driving dataset.

**Summary Of The Review:**

I value some of the contributions of the paper such as adapting StyleGAN3 to video generation. However, I have multiple concerns regarding the experimental results as well as the conclusions drawn from them. I'm happy to hear from the authors about their thoughts on these concerns and reconsider my rating.

---

> ### Author Response · Authors · 2022-11-15
> **Response to Reviewer yrRK (1/2)**
>
> **Q1: Only two landscape video datasets are used to evaluate the method.**
>
> A1: Following the suggestion, we evaluate our approach on a new dataset, Taichi-HD [1]. In particular, on this dataset, we both conduct ablation studies to investigate the function of each proposed component, and compare our approach to existing methods to verify our superiority. The results are listed below and also *updated in Tab. 2 and Tab. 3 in the submission*. We can tell that the conclusions obtained from this dataset are consistent with those from SkyTimelapse and YouTube Driving.
>
> |                    | $FID$ | $FVD_{16}$ | $FVD_{128}$ |
> |--------------------|-----|-----|-----|
> | A StyleGAN-V       | 33.8 | 152.0 | 267.3 |
> | B + Alias free     | 31.4 | 171.7 | 522.9 |
> | C + Image pretrain | **20.8** | 104.3 | 314.2 |
> | D + TSM            | 26.0 | 84.6  | 176.2 |
> | E + B-Spline       | 25.5 | **82.6**  | **169.7** |
> | F + Low-rank       | 24.8 | 98.3  | 186.8 |
>
> **Q2:  The trends of the results are not well aligned.**
>
> A2:  This might be a misunderstanding caused by our result organization.
>
> First, the introduction of image pre-training (Config-C) targets improving the per-frame quality after using the alias-free technique. In this case, we focus more on the FID metric. We transcribe a part of Tab. 1 as the table below, where we can see that Config-C outperforms Config-B regarding FID on *all three* datasets.
>
> |          | SkyTimelapse | YouTube Driving | Taichi-HD |
> |----------|--------------|-----------------|-----------|
> | Config-B | 54.0         | 56.4            | 31.4      |
> | Config-C | **52.2**         | **15.6**            | **20.8**      |
>
> Similarly, the introduction of TSM aims to learn more reasonable dynamics, which are usually evaluated with $FVD_{16}$ and $FVD_{128}$. We transcribe the results regarding these two metrics in the table below, we can see that Config-D beats Config-C on *all three* datasets.
>
> |    $FVD_{16}$   | SkyTimelapse | YouTube Driving | Taichi-HD |
> |----------|--------------|-----------------|-----------|
> | Config-C | 73.5         | 272.8           | 104.3     |
> | Config-D | **49.0**         | **207.2**           | **84.6**      |
>
> |    $FVD_{128}$      | SkyTimelapse | YouTube Driving | Taichi-HD |
> |----------|--------------|-----------------|-----------|
> | Config-C | 230.3         | 447.5           | 314.2     |
> | Config-D | **135.9**         | **221.5**           | **176.2**      |
>
> We *have updated the caption of Tab. 1* to facilitate a better interpretation of this table.
>
> **Q3: Recently published methods should be considered.**
>
> A3: Following the suggestion,  we compare our approach against recent approaches [2,3]. The results on SkyTimelapse and Taichi-HD are shown below, where we surpass the other two approaches regarding both $FVD_{16}$ and $FVD_{128}$. We *have also updated the results in Tab. 2 and Tab. 3*.
>
> |   SkyTimelapse           | $FVD_{16}$ | $FVD_{128}$ |
> |--------------|-------|--------|
> | TATS-base [2]    | 132.6 | -      |
> | LongVideoGAN [3] | 116.5 | 152.7  |
> | Ours         | **49.0**  | **135.9**  |
>
> |   Taichi-HD |$FVD_{16}$ | $FVD_{128}$ |
> |--------------|-------|--------|
> | TATS-base [2]    | 94.6 | -      |
> | Ours         | **84.6**  | 176.2  |
>
> **Q4: More metrics to show the advantage of the proposed method (Config E and F).**
>
> A4: Thanks. In fact, existing quantitative metrics, such as $FVD_{16}$ and $FVD_{128}$, usually struggle in capturing long-range dynamics and hence may not be reliable enough for the evaluation of long video generation. Following the suggestion, we conduct a user study (with 50 annotators) on the output videos regarding different lengths of time, including single frames, short videos (128 frames), and long videos (500 frames). As shown in the table below, Config-F received more votes than Config-D in terms of long video generation. The above discussion, together with the table below, *has been included in the appendix (see Sec. C)*. We *have also included a qualitative comparison between Config-E and Config-F in the appendix (see Fig. A6)* to verify that our approach helps alleviate the issue of generating cyclic videos.
>
> |          | YouTube Driving |      |    |   Taichi-HD  |             |            |
> |:--------:|:---------------:|:-----------:|:----------:|:------------:|:-----------:|:----------:|
> |          |   Single Image  | Short Video | Long Video | Single Image | Short Video | Long Video |
> | Baseline |        4        |      6      |      8     |       0      |      2      |      4     |
> | Config-D |        **62**       |     **60**     |     12     |      **52**      |      **66**     |     36     |
> | Config-F |        34       |      34     |     **80**     |      48      |      32     |     **60**     |
>
> [1] First order motion model for image animation. NIPS 2019
>
> [2] Long video generation with time-agnostic vqgan and time-sensitive transformer. ECCV 2022.
>
> [3] Generating long videos of dynamic scenes. NeurIPS 2022.

---

> > ### Author Response · Authors · 2022-11-15
> > **Response to Reviewer yrRK (2/2)**
> >
> > **Q5: When pretraining the model on images, what is the input as the motion embedding?**
> >
> > A5: Motion embeddings are randomly sampled. They are further processed by Conv1D operation. (However, during the image pretraining stage, the Conv1D operation will not receive meaningful updates.)
> >
> > **Q6: When loading the pretrained parameters to the model, are the shallow layers initialized and trained from scratch?**
> >
> > A6: We load the weights of both shallow layers and deep layers, and perform fine-tuning on all parameters.
> >
> > **Q7: When generating an infinite length of video, what happens if $t$ is sampled to be larger than $N_A$ - 1?**
> >
> > A7: If $t$ is too large, we simply divide it into small intervals. For each interval, a fixed number ($N_A$) of anchors are instantiated. After each chunk’s generation, we simply concatenate all the generated short clips to form the final video with length $t$.
> >
> > **Q8: Which layers are TSM added?**
> >
> >
> > A8: TSM is added to each layer before the temporal fusion function. We *have updated Eq. (7)* to clarify this.
> >
> > **Q9: Reproducibility?**
> >
> > A9: The code and models will be made publicly available.

---

### Official Review · Reviewer_pmsF · 2022-10-24

**Confidence:** 4
**Correctness:** 3
**Technical Novelty And Significance:** 3
**Empirical Novelty And Significance:** 3
**Recommendation:** 6

**Clarity, Quality, Novelty And Reproducibility:**

- It is not clear what “our method” refers to among the Config-B - F. Is it Config-D for short videos, and Config-F for long videos?

- It would be also better to have a name for the proposed method instead of Config-A, B, …

**Strength And Weaknesses:**

Strength
+ They propose multiple techniques to siginficantly improve the video synthesis quality. Especially the FVD scores improved much compared to the prior arts.

+ It resolves the brush effect which has been a problem in the previous methods, due to the generated contents being bound to the spatial coordinates in the image/frame.

+ The visual quality of the generated video outperforms those of the previous methods.

+ Long video generation is also considered, and is come up with novel methods to handle this.

Weakness
- While the proposed multiple techniques do improve over the baseline, the conclusion from the result is not very clear. In Table 1. Config-D shows the best FVD scores on the two datasets but not the methods with the best FID score are different over these two datasets. Why is the Config-E and F worse than D? What is the best Config for users?

- While the paper argues the proposed method does not sacrifice the per-frame quality, the FID scores are worse than the baseline StyleGan-V.

- The visual comparison (Figure 2, 3, A4, A5) shows the results on different sequence examples, which makes it hard to directly compare the performance between the baseline vs proposed method.





**Summary Of The Paper:**

This work proposes a method for smooth video synthesis. They build upon the SyleGan-V and progressively improve the baseline by proposing the alias-free operation, adding a temporal shift module, and incorporating the B-Spline based motion representation. It achieves improved temporal smoothness in the generated video without hurting the per-frame quality much. The experiments on SkyTimelapse and YouTube Driving datasets demonstrate the contributions of each component of the method. This work also considers long video generation which requires continuously sampling the temporal latent codes.

**Summary Of The Review:**

The discussion on the worse per-frame FID score, and visual comparison on the same examples, and the best method among the Configs for short / long video will make this paper more complete.

---

> ### Author Response · Authors · 2022-11-15
> **Response to Reviewer pmsF**
>
> **Q1: While the proposed multiple techniques do improve over the baseline, the conclusion from the result is not very clear. In Table 1. Config-D shows the best FVD scores on the two datasets but not the methods with the best FID score are different over these two datasets. Why is Config-E and F worse than D? What is the best Config for users?**
>
> A1: First of all, we would like to state the conclusion, which is that Config-D gives the best performance for short video generation and Config-F most benefits long video generation. Users can choose their favorable configuration depending on the needs. We *have added the above suggestion at the end of Sec. 2*.
>
> Then, we try to address your concerns with a detailed explanation. The metrics used in Tab. 1 help evaluate the quality of single frames (via FID) and short videos (via FVD) respectively, where Config-D performs best. When it comes to long video generation (*e.g.* generating 500 frames), existing metrics, such as $FVD_{16}$ and $FVD_{128}$, usually struggle in capturing long-range dynamics and hence may not be reliable enough. We hence include a user study (with 50 annotators) on the output videos regarding different lengths of time, including single frames, short videos (128 frames), and long videos (500 frames), as an additional support. As shown in the table below, Config-F received more votes than Config-D in terms of long video generation. The above discussion, together with the table below, *has been included in the appendix (see Sec. C)*.
> |          | YouTube Driving |             |            |   Taichi-HD  |             |            |
> |:--------:|:---------------:|:-----------:|:----------:|:------------:|:-----------:|:----------:|
> |          |   Single Image  | Short Video | Long Video | Single Image | Short Video | Long Video |
> | Baseline |        4        |      6      |      8     |       0      |      2      |      4     |
> | Config-D |        **62**       |     **60**     |     12     |      **52**      |      **66**     |     36     |
> | Config-F |        34       |      34     |     **80**     |      48      |      32     |     **60**     |
>
> **Q2: While the paper argues the proposed method does not sacrifice the per-frame quality, the FID scores are worse than the baseline StyleGan-V.**
>
> A2: First, we would like to remind that, instead of per-frame quality that is usually evaluated by FID, video generation cares more about the soundness of a sequence of frames, which is usually evaluated by FVD. Regarding the FVD results, our approach achieves the best performance on *all three* datasets, as shown in Tab. 1. As for the FID results, our approach surpasses StyleGAN-V on YouTube Driving and Taichi-HD but is worse on SkyTimelapse. Following the suggestion, we *have changed the claim* to “without sacrificing per-frame quality too much” to be more accurate.
>
> **Q3: The visual comparison (Figure 2, 3, A4, A5) shows the results on different sequence examples, which makes it hard to directly compare the performance between the baseline vs proposed method.**
>
> A3: All samples shown in this paper are randomly generated. It is almost unachievable to control two independently learned models (such as the baseline and our different configurations) to produce the same video contents.
>
> **Q4: It is not clear what “our method” refers to among the Config-B - F. Is it Config-D for short videos, and Config-F for long videos?**
>
> A4: If not specified, our method refers to Config-D for short video generation and to Config-F for long video generation, which are the best configurations for these two settings. We *have added this statement at the end of Sec. 2* to make this clear.

---

### Official Review · Reviewer_xq3D · 2022-10-27

**Confidence:** 4
**Correctness:** 3
**Technical Novelty And Significance:** 3
**Empirical Novelty And Significance:** 3
**Recommendation:** 6

**Clarity, Quality, Novelty And Reproducibility:**

The paper is well written. The backgrounds are well-covered. The paper included comprehensive quantitative evaluations but qualitative ones are lacking. The techniques used in the paper are not original. However, it is inspirational to see that these techniques can be used together to obtain a more powerful model. The authors promised in the paper to release code.

**Strength And Weaknesses:**

## Strength
* The proposed method significantly improves upon the previous works in terms of video quality. What's more important, the paper conveyed a message that good quantitative scores does not always equal good visual quality perceived by humans.
* The paper is easy to follow, and includes comprehensive quantitative experiments and good visualization.

## Weaknesses
* Most of the techniques used in the paper are not new. They are borrowed from related works.
* The effect of setting F is not clear. There is no figures or videos showing what referred to as *similar content periodically appears*.
* The new techniques introduced in the paper (setting E and F) significantly hurt the quantitative performance. Although claimed to have the benefit of reducing the jitter artifact, the non-visual evidence is weak (Table 4). It will be more persuasive if there are better evaluations, such as a user study.

**Summary Of The Paper:**

The paper proposed a GAN-based video generative model that is based upon StyleGAN-V while producing videos with significantly fewer artifacts. The paper first outlined several common artifacts seen in the videos produced by StyleGAN-V, and then provided an extensive set of targeted solutions:
1. Alias-free techniques from StyleGAN3 is adopted to reduce the texture sticking artifacts at the cost of a slight drop in quantitative scores.
2. To alleviate the quality drop, the author proposed pretraining the later layers on a image generation task and then finetuning the early layers on the actual video generation task.
3. A *temporal shift module*, originally developed for video understanding, is incorporated into the discriminator, improving the quality further.
4. To enable jitter-free long video generation, the motion latent space is instead modeled with a B-Spline.
5. To reduce the periodic repeating artifact introduced by 4, a low-rank matrix is used to limit the effect of temporal features on the output frames.

In terms of quantitative results, the proposed method with improvement 1, 2, and 3 achieved the best video quality on an existing benchmark but inferior image quality. It however outperformed prior art in every aspect on a new, self-collected dataset.

Adding improvement 4 and 5 will improve visual quality but reduce quantitative scores.

**Summary Of The Review:**

I liked how the paper cast light on the practical issues in the previous works and then tacked them one-by-one, instead of focusing only on getting better numerical scores. This, as well as the techniques proposed in the paper, can serve as a good reference point for the future works. There are a few improvements that are not as well-justified as others, but it does not diminish other contributions.

---

> ### Author Response · Authors · 2022-11-15
> **Response to Reviewer xq3D**
>
> **Q1: Most of the techniques used in the paper are not new. They are borrowed from related works.**
>
> A1: The techniques, such as TSM and B-Spline, are originally designed for other vision tasks. In this work, based on the analysis of some challenges in video generation, we show, *for the first time*, that these techniques can facilitate smooth long video generation in an ingenious way. We believe that, with these newly introduced components, our approach serves as a simple yet strong baseline for video generation, and also points out some directions that are worth further exploration.
>
> **Q2: The effect of setting F is not clear. There are no figures or videos showing what referred to as similar content periodically appears.**
>
> A2: We *have added a new figure in the appendix (see Fig. A6)* to help understand the issue of “similar content periodically appearing”. As shown in Fig. A6a, the 300th frame, 600th frame, and 900th frame generated by Config-E tend to have similar scene layouts and objects. Instead, Config-F in Fig. A6b can produce a sequence of frames with more diverse contents.
>
> **Q3: The new techniques introduced in the paper (setting E and F) significantly hurt the quantitative performance. Although claimed to have the benefit of reducing the jitter artifact, the non-visual evidence is weak (Table 4). It will be more persuasive if there are better evaluations, such as a user study.**
>
> A3: Thanks. In fact, Config-E and Config-F are particularly designed to address issues of jittering and “periodic visual contents” in the case of long video generation. Existing evaluation metrics, such as $FVD_{16}$ and $FVD_{128}$, usually struggle in capturing long-range dynamics and hence may not be reliable enough. Following the suggestion, we conduct a user study (with 50 annotators) on the output videos regarding different lengths of time, including single frames, short videos (128 frames), and long videos (500 frames). As shown in the table below, Config-D performs well from the perspectives of generating single frames and short videos, Config-F performs far better for long video generation. The above discussion, together with the table below, *has been included in the appendix (see Sec. C)*.
> |          | YouTube Driving  |             |           |   Taichi-HD  |             |            |
> |:--------:|:---------------:|:-----------:|:----------:|:------------:|:-----------:|:----------:|
> |          |   Single Image  | Short Video | Long Video | Single Image | Short Video | Long Video |
> | Baseline |        4        |      6      |      8     |       0      |      2      |      4     |
> | Config-D |        **62**       |      **60**     |     12     |     **52**      |     **66**     |     36     |
> | Config-F |        34       |      34     |     **80**     |      48      |      32     |     **60**     |

---

> > ### Comment · Reviewer_xq3D · 2022-12-07
> > **Thanks for the response**
> >
> > Dear authors,
> >
> > Thanks for the response and the extra user study. This resolves my concerns around lacking of evidences to justify config E and F. Although the proposed method does not triumph on everything, I agree that it has made useful contributions.
> >
> > Best

---

### Author Response · Authors · 2022-11-25
**Any further comment?**

Dear reviewers and AC

Thanks a lot for your effort in reviewing this submission! We have tried our best to address the mentioned concerns/problems in the rebuttal. Feel free to let us know if there is anything unclear or so. We are happy to clarify them.

Best, Authors

---

### Decision · Program_Chairs · 2023-01-20

**Decision:**

Accept: poster

**Justification For Why Not Higher Score:**

Reviewers agreed that the results were an improvement over StyleGAN-V and the paper was ok to accept, but no reviewers were particularly enthusiastic about the paper; the results are better than StyleGAN-V but not dramatically so, and the technical ideas were mostly adapted from other problems rather than presenting some dramatic new techniques.

**Justification For Why Not Lower Score:**

All reviewers recommended acceptance.

**Metareview: Summary, Strengths And Weaknesses:**

The paper presents a generative model for videos built on StyleGAN-V. In initial reviews all reviewers found the paper to have adequate contributions, thought that it was mostly well-written, and presented results which were a clear improvement over StyleGAN. Several reviewers were confused about Configurations E and F. Reviewer yrRK also pointed out several other important weaknesses: the paper only evaluated on two datasets of landscape videos, and should have compared with some more recent prior work. These issues were mostly resolved by the author's responses, and in the end all reviewers agreed that the paper should be accepted.

**Note From Pc:**

if the above contains the word "oral" or "spotlight" please see: "oral" presentation means -> notable-top-5% and "spotlight" means -> notable-top-25%. As stated in our emails, we are disassociating presentation type from AC recommendations